# Understanding maternal sepsis risk factors and bacterial etiology: A case control study protocol

**Kelly Thompson** [1,2]*, **Duy Pham Thanh**[3], **Jane E. Hirst**[4], **Mark Woodward**[1,4], **Hai Pham Thanh**[5], **Huong Tran Thi Lien**[5], **Kiet Tao Tuan**[5], **Binh Le Thanh**[5], **Evelyne Kestelyn** [3], **Thuan Dang Trong**[3], **Katie Harris**[1], **Linh Nguyen Thi My**[3], **Hien Vu Thi Minh**[3], **Tuyen Ha Thanh**[3], **Thanh Le Quang**[5], **Louise Thwaites**[3]

1 The George Institute for Global Health, University of New South Wales, Sydney, Australia, 2 Nepean Blue Mountains Local Health District, Kingswood, Australia, 3 Oxford University Clinical Research Unit, Ho Chi Minh City, Vietnam, 4 The George Institute for Global Health, Imperial College London, London, United Kingdom, 5 Tu Du Hospital, Ho Chi Minh City, Vietnam

* kthompson@georgeinstitute.org

**Data Availability Statement:** No datasets were generated or analysed during the current study. All relevant data from this study will be made available upon study completion.

## Abstract

### Introduction

Maternal disorders are the third leading cause of sepsis globally, accounting for 5.7 million (12%) cases in 2017. There are increasing concerns about the emergence of antimicrobial resistance (AMR) in bacteria commonly causing maternal sepsis. Our aim is to describe the protocol for a clinical and microbiology laboratory study to understand risk factors for and the bacterial etiology of maternal sepsis in a tertiary Obstetrics and Gynaecology Hospital.

### Methods

This case-control study aims to recruit 100 cases and 200 controls at Tu Du Hospital in Ho Chi Minh City, Vietnam, which had approximately 55,000 births in 2022. Women aged $\geq$ 18 years and $\geq$ 28 weeks gestation having a singleton birth will be eligible for inclusion as cases or controls, unless they have an uncomplicated localised or chronic infection, or an infection with SARS-CoV-2. Cases will include pregnant or recently pregnant women with sepsis recognised between the onset of labour and/or time of delivery/cessation of pregnancy for up to 42 days post-partum. Sepsis will be defined as suspected or confirmed infection with an obstetrically modified Sequential Organ Failure Assessment score of $\geq$ 2, treatment with intravenous antimicrobials and requested cultures of any bodily fluid. Controls will be matched by age, location, parity, mode of delivery and gestational age. Primary and secondary outcomes are risk factors associated with the development of maternal sepsis, the frequency of adverse outcomes due to maternal sepsis, bacterial etiology and AMR profiles of cases and controls.

**Funding:** This paper was funded by the National Health and Medical Research Council of Australia, Investigator Grant received by Dr. Kelly Thompson (APP1194058). Partial funding was received from Maridulu Budyari Gamal, the Sydney Partnership for Health, Education, Research and Enterprise (SPHERE), received by Drs. Kelly Thompson, Jane Hirst and Louise Thwaites. The funders had no role in study design, data collection and analysis, decision to publish, or preparation of the manuscript.

**Competing interests:** The authors have declared that no competing interests exist.

## Discussion

This study will improve understanding of the epidemiology and clinical implications of maternal sepsis management including the presence of AMR in women giving birth in Vietnam. It will help us to determine whether women in this setting are receiving optimal care and to identify opportunities for improvement.

## Introduction

Maternal sepsis is life-threating organ dysfunction that occurs in response to infection during pregnancy, childbirth, post-abortion or in the postpartum period [1]. Maternal disorders are the third leading cause of sepsis globally, accounting for 5.7 million (12%) cases in 2017 [2]. Despite a lack of data, particularly from low-income and middle-income country settings (LMICs), an estimated 7% of pregnant or post-partum women will develop an infection requiring hospital treatment [3], with sepsis responsible for one in 10 maternal deaths globally [4–6].

Efforts to address maternal sepsis have been largely integrated into broader initiatives to reduce maternal and neonatal mortality [7]. A major emphasis of programs has been on increasing the number of births with a skilled attendant, particularly within health facilities. It is estimated that three-quarters of all births globally now occur within health facilities, but as this number has increased, concerns regarding the quality of maternity care in health facilities have emerged [7].

Although antibiotics unquestionably save lives in childbirth [8], they should always be used in the context of antimicrobial stewardship programmes [7,9]. Bacterial characterisation and antimicrobial susceptibility testing are important in determining appropriate treatment, particularly in LMICs where high rates of antimicrobial resistance (AMR) present one a challenge to global public health [9,10]. Emerging concerns about antimicrobial resistance (AMR) in bacteria commonly causing maternal sepsis [7,8] are polarised by an absence of data on antimicrobial stewardship and resistance in maternity units globally [7,8].

Reducing maternal mortality and morbidity due to sepsis requires prevention, early diagnosis and prompt but appropriate management. Several interventions have been designed to prevent and promptly recognise sepsis, including early warning systems and care bundles. However, data describing how maternal sepsis is recognised in urban health care facilities are lacking [3], and there are no validated criteria for diagnosis.

Vietnam is the third largest LMIC in Southeast Asia with a population of close to 98.2 million [11]. Like other parts of Southeast Asia, Vietnam is a hotspot for emerging infectious diseases with severe dengue, Streptococcus and increased AMR listed as major causes of sepsis admissions in intensive care units [12]. While there is some evidence describing the country-specific etiology of sepsis and septic shock in Vietnam, these reports are limited to adults [13] and neonates [10,14] requiring intensive care treatment. There are limited contemporary data on risk factors for and causes of sepsis, including the emergence of antimicrobial resistant bacteria in pregnant women and women who have recently given birth [10].

Our aim is to describe the protocol for a clinical and microbiology laboratory study to understand the epidemiology of maternal sepsis including risk factors, current use of antimicrobial agents and stewardship practices and bacterial etiology (pathogen profiles and AMR) in a large tertiary Obstetrics and Gynaecology Hospital in Ho Chi Minh City, Vietnam.

## Materials and methods

### Study design, setting and population

This case-control study will be conducted in Tu Du Hospital in Ho Chi Minh City located in the Southern region of Vietnam. Tu Du Hospital is a specialised obstetrics and gynaecological tertiary referral centre and one of the largest hospitals in Asia, with approximately 55,000 births in 2022.

### Inclusion criteria—cases

Cases are defined as pregnant or recently pregnant women with sepsis recognised between the onset of labour and/or time of delivery/cessation of pregnancy for up to 42 days post-partum. A diagnosis of sepsis will require the woman to have:

- A suspected or confirmed infection being treated with intravenous antibiotics

- An Obstetrically Modified Sequential Organ Failure Assessment (SOFA) Score of $\geq 2$ (Table 1)

- Requested cultures of any bodily fluid, including swab specimens

### Inclusion criteria—controls

Controls will include pregnant or recently pregnant women who are up to 42 days post-partum who do not have sepsis.

Controls will be matched by age (within 2-year age bands), location (urban vs rural), mode of delivery (normal vaginal delivery or caesarean section), parity (primiparous vs multiparous) and gestational age rounded to the closest week (where possible). For each case, two controls will be randomly selected to ensure the comparability of the controls to the general obstetric population.

### Case and control exclusion criteria

1. Women aged < 18 years of age

2. Women who are < 28 weeks gestation

3. Women who have an uncomplicated localised or chronic infection

4. Women who have an infection with SARS-CoV-2 (COVID-19)

**Table 1. Obstetrically modified SOFA score [15].**

| Clinical Parameter | Score | | |
| --- | --- | --- | --- |
| | 0 | 1 | 2 |
| Respiratory: $SPO_2/FiO_2$ (mmHg) [16] | $\geq 512$ | 357–511 | <357 |
| Coagulation: Platelets x$10^6$/L | $\geq 150$ | 100–149 | <100 |
| Liver: Bilirubin (μmol/L) | $\leq 20$ | 21–32 | >32 |
| Cardiovascular: Mean Arterial Pressure (MAP) (mmHg) | MAP$\geq 70$ | MAP<70 | Vasopressors |
| Central Nervous System | Alert | Rousable by voice | Rousable by pain |
| Renal: Creatinine (μmol/L) | $\leq 90$ | 91–120 | >120 |

Abbreviations: $SPO_2$ = oxygen saturation; $FiO_2$ = fraction of inspired oxygen; μmol/L = micro moles per litre; mmHg = millimetres of mercury.

## Screening

Women with a suspected or confirmed infection will be assessed for inclusion into the study by screening for possible markers of organ failure using the Obstetrically Modified quick Sequential Organ Failure Assessment (OMqSOFA) score [15]. The OMqSOFA scores one point for each of the following criteria (respiratory rate $\geq 25$ breaths per minute; systolic blood pressure <90mmHg; any non-alert mental status). A score of 2 or more indicates a higher risk of infection and sepsis and prompts further screening.

For women who have an OMqSOFA score of $\geq 2$ points, organ failure screening will be conducted using the Obstetrically Modified SOFA score (Table 1). A diagnosis of maternal sepsis will require the woman to have a suspected or confirmed infection plus an Obstetrically Modified SOFA score of $\geq 2$. The Obstetrically Modified SOFA score has been used in accordance with recommendations from the Society of Obstetric Medicine in Australia and New Zealand [15]. In order to demonstrate evidence of organ dysfunction a score of $\geq 2$ is required [15]. To simplify screening, we removed all individual organ failure scores of 3 and 4 from the screening sheet, as an overall score of $\geq 2$ across all organ systems is required for study inclusion. Modified SOFA parameters for pregnant or recently pregnant woman include a reduction in the creatinine threshold, as serum creatinine levels are significantly reduced in pregnancy with normal ranges of 35–80 μmol/L and a simplified neurological assessment as Glasgow Coma Score is not routinely assessed on maternity wards. For this study, we also modified the respiratory SOFA score component, substituting the $PaO_2/FiO_2$ with the $SPO_2/FiO_2$ ratio [16], due to lack of availability of arterial blood gas monitoring in the hospital.

## Outcomes

The primary outcome is to identify risk factors associated with the development of maternal sepsis. Secondary outcomes include the frequency of adverse outcomes due to maternal sepsis, including death and treatment with invasive organ support, bacterial etiology and their AMR profiles comparing cases and controls. Timely and appropriate management of maternal sepsis including the time to blood cultures, antibiotic administration, primary antimicrobial use, dose and duration of antimicrobial treatment, perinatal outcomes including complications in pregnancy and neonatal outcomes will also be assessed.

Clinical samples will be collected and subjected to microbiological culture in accordance with routine clinical care and procedures. Where clinically indicated for patient management vaginal swabs will be collected. Bacterial identification and AMR testing will be performed in-house using the BD Phoenix™ automated identification and susceptibility testing system. Bacterial isolates identified from sepsis cases will undergo bacterial and phenotypic confirmatory tests and molecular investigation (AMR gene typing, genome sequencing). For controls, two clinical samples in addition to standard care (vaginal and rectal swabs) will be taken. Samples from controls will undergo bacterial identification and AMR testing. In the control population we will assess colonising bacteria in vaginal and rectal swabs e.g. Group A, B, C and G *Streptococcus*, *Escherichia coli*, *Enterobacter spp*, *enterococcus*, *Chlamydia trachomatis* and *staphylococcus aureus*. These colonising bacteria will be subjected to whole-genome sequencing. To better understand what causes maternal sepsis we will assess the genetic relatedness of the same bacterial species identified from cases and controls.

## Statistical analysis plan

Data from the case-control study will be cleaned and a descriptive analysis conducted. Approximately symmetric continuous variables will be reported as mean ± standard deviation. Skewed continuous variables will be reported as median and interquartile range (IQR).

Categorical data will be reported as proportions (%). Outcomes will be censored at the end of the postpartum period (42 days) or at discharge, transfer to another treating facility or at time of death, whichever comes first.

Risk factors associated with sepsis will be compared between cases and controls. Risk factors will include sociodemographic, pregnancy, labour and delivery characteristics. Due to the uneven case-control distribution, and also due to the matched allocations, conditional logistic regression models will be used to compare the actions of putative risk factors between the cases and controls. We will assess individual risk factors, including but not limited to the mothers baseline sociodemographic and physiological pregnancy characteristics. Statistical adjustment will be made for potential confounders such as the presence of anaemia and other complications of pregnancy (hypertensive disorders and gestational diabetes) as well as matching variables. A two-sided p-value of <0.05 will be considered statistically significant. Analysis will be conducted using Stata software.

Bacterial isolates and antimicrobial resistance profiles identified via microbiological cultures will be reported and compared between cases and controls. Bacterial and phenotypic confirmatory tests, molecular investigation (AMR gene typing, whole-genome sequencing) of major infecting organisms will be compared to commensal organisms collected from controls.

We chose to study 100 cases and 200 controls primarily because of the anticipated availability of suitable cases and controls within the limitations of the research environment. With this sampling allocation we will have 80% power, at the 5% significance level and using a two-sided test, to detect an odds ratio of 3, or less, whenever the risk factor has a prevalence of 8%, or more.

## Data management

Data management will be undertaken by the Oxford University Clinical Research Unit (OUCRU) in Ho Chi Minh City. Data will be collected on paper-based Case Report Forms at site and transcribed to the electronic study database (CliRes data management system) by the study coordinators and hosted at OUCRU. A separate master log of enrolments containing identifying data will be stored securely in a password protected excel file by the study coordinator on site. This will be referred to by personnel at the study site to address questions of data integrity or to verify data, as requested by the coordinating centre.

A copy of the Case Report Form is provided in the Supporting Information. Data collection is broadly grouped into the following categories; women's sociodemographic characteristics, relevant medical history, labour and delivery details, including complications, clinical identification of infection/sepsis and treatment, laboratory investigations, ICU data collection for those admitted to ICU with infection/sepsis, length of stay outcomes, Edinburgh depression score at 42 days and infant outcomes.

## Ethical considerations

The study is approved by Oxford Tropical Research Ethics Committee (approval number: 501–22) and the Ethics Committee of the Tu Du hospital (approval number: 2979/BVTD-HDDD). This is a low-risk research project, where the only foreseeable risk is one of discomfort. The site Principal Investigator will seek written informed consent from patients (or a delegated legal representative where appropriate) prior to enrolment into the study. All patient information sheets and consent forms will be written in Vietnamese.

## Study status and timeline

A planned one year recruitment period commenced in September 2023, with termination of recruitment after 100 cases and 200 controls are enrolled.

## Discussion

Maternal sepsis reamins a health condition with limited reseach focus in Vietnam, resulting in notable knowledge gaps in our understanding of disease epidemiology, treatment attributes and associated outcomes. This study seeks to enhance our understanding of the clinical implications of maternal sepsis, its management and the presence of AMR in women giving birth in a large teaching hospital. By gaining these insights, we aim to determine whether women in this setting are receiving optimal care and to identify gaps and opportunities to improve care. This study was conceived in 2020 as a collaborative multidisciplinary project across several research and clinical disciplines (epidemiology, public health, molecular biology, statistics, obstetrics, gyanecology, intensive care and infectious diseases) and commenced recruitment in September 2023.

The main limitation is the single centre nature of the study which will limit the generalisability of findings. However, the findings will be relevant for the hospital where the research is being conducted and given the large number of births annually, may lead to improvements in prompt recogition and early initiation of treatment in local sepsis cases. The second limitation relates to the use of case-control methodology. However, as data on this topic are scarce this was considered appropriate to achieve the required number of maternal sepsis cases and to better understand the context of maternal sepsis recognition, treatment and management and AMR in this clinical setting, before designing a prospective or interventional study.

The results of the study will be used to inform changes to clinical practie in the study setting (as required) and will be published in a peer-reviewed journal and presented at national and international conferences.

## Supporting information

**S1 Checklist.**
(DOCX)

**S1 File. Case report form.**
(DOCX)

## Acknowledgments

Tu Du Hospital Staff Role
 Phan Thị Thủy Head Nurse Ward A
 Nguyễn Thị Kim Bình Head Nurse Post-operative
 Nguyễn Thị Thêm Head Nurse Ward H
 Bùi Kim Chi Head Nurse Ward N1
 Nguyễn Thị Hồng Phượng Head Nurse Ward N2
 Trần Ngọc Mỹ Head Nurse Ward M
 Dr. Mã Thanh Tùng Vice head-Department of anesthesia and resuscitation
 Dr. Trần Thị Hồng Vân Department of anesthesia and resuscitation
 Dr. Huỳnh Công Trung Department of anesthesia and resuscitation
 Dr. Phan Thị Thắm Department of anesthesia and resuscitation
 Dr. Nguyễn Đỗ Tiền Department of anesthesia and resuscitation
 Ngô Đức Toàn Department of anesthesia and resuscitation
 Lê Chí Hiếu Department of anesthesia and resuscitation
 Huỳnh Thị Xuân Lan Department of anesthesia and resuscitation
 Dr. Lê Minh Hoài An Head of Laboratory Department
 Dr. Trần Vũ Hòa Vice-head of Laboratory Department

## Author Contributions

**Conceptualization:** Kelly Thompson, Duy Pham Thanh, Jane E. Hirst, Louise Thwaites.

**Methodology:** Kelly Thompson, Duy Pham Thanh, Mark Woodward, Katie Harris, Thanh Le Quang, Louise Thwaites.

**Project administration:** Kelly Thompson, Duy Pham Thanh, Hai Pham Thanh, Huong Tran Thi Lien, Kiet Tao Tuan, Binh Le Thanh, Evelyne Kestelyn, Thuan Dang Trong, Linh Nguyen Thi My, Hien Vu Thi Minh, Thanh Le Quang, Louise Thwaites.

**Writing – original draft:** Kelly Thompson, Louise Thwaites.

**Writing – review & editing:** Kelly Thompson, Jane E. Hirst, Mark Woodward, Hai Pham Thanh, Huong Tran Thi Lien, Kiet Tao Tuan, Binh Le Thanh, Thuan Dang Trong, Katie Harris, Linh Nguyen Thi My, Tuyen Ha Thanh, Louise Thwaites.

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
