## [Decision Letter · Decision Letter 0]

4 Mar 2024

PONE-D-23-40494Understanding maternal sepsis risk factors and bacterial etiology: a case-control study protocolPLOS ONE

Dear Dr. Thompson,

Thank you for submitting your manuscript to PLOS ONE. After careful consideration, we feel that it has merit but does not fully meet PLOS ONE’s publication criteria as it currently stands. Therefore, we invite you to submit a revised version of the manuscript that addresses the points raised during the review process.

We look forward to receiving your revised manuscript.

Kind regards,

Trung Quang Nguyen

Academic Editor

PLOS ONE

 [National Health and Medical Research Council of Australia, Investigator Grant APP1194058 ].  

Reviewers' comments:

Reviewer's Responses to Questions

**Comments to the Author**

1. Does the manuscript provide a valid rationale for the proposed study, with clearly identified and justified research questions?

Reviewer #1: Partly

Reviewer #2: Yes

Reviewer #3: Partly

Reviewer #4: Yes

2. Is the protocol technically sound and planned in a manner that will lead to a meaningful outcome and allow testing the stated hypotheses?

Reviewer #1: Partly

Reviewer #2: Yes

Reviewer #3: Yes

Reviewer #4: Partly

3. Is the methodology feasible and described in sufficient detail to allow the work to be replicable?

Reviewer #1: Yes

Reviewer #2: Yes

Reviewer #3: Yes

Reviewer #4: Yes

4. Have the authors described where all data underlying the findings will be made available when the study is complete?

Reviewer #1: Yes

Reviewer #2: Yes

Reviewer #3: Yes

Reviewer #4: Yes

5. Is the manuscript presented in an intelligible fashion and written in standard English?

Reviewer #1: No

Reviewer #2: Yes

Reviewer #3: Yes

Reviewer #4: Yes

6. Review Comments to the Author

You may also provide optional suggestions and comments to authors that they might find helpful in planning their study.

Reviewer #1: Understanding maternal sepsis risk factors and bacterial etiology: a case-control study protocol

Reviewer Comments:

Keyword:

Antimicrobial resistance should not be in the keyword as it’s not in line with title of the protocol

Abstract:

Line 30 contain was thus grammatical error. A protocol can’t contain was. Overall grammatical errors should be rewrite.

Line 43 primary and secondary outcome should be mentioned clearly in the abstract section

Overall aim and title conflict rigorously.

Introduction:

Line 56 LMICs should be showed a grammatical mistake.

Line 63, 64 should contain background data more to prove the statement as well.

Line 65 referencing should be corrected [7] [8]

More epidemiology on bacterial profile and AMR should be reflected in accordance with the study objective of the following concern study.

Material and Methods:

Line 82 how cases and controls will be selected not clear. If same age and gestation will be treated as either case or control? Line 93 conflict with it without defining pregnant of which gestational age.

Line 97 mode of delivery should be specified by NVD or CS

The Table -1, Obstetrically Modified SOFA score according to the reference [11] was not mentioned why respiratory parameter, coagulation and liver parameter was changed was not described and reference should be mentioned for the justified modification.

In line 117-118 how Bacterial etiology and AMR profile will be assessed through transmission dynamics or genomic analysis (16s r RNA extraction or shotgun metagenomics in which region v4 or any other should be mentioned clearly.

The overall outcome should be focused and specified according to the research question and hypothesis that lack the specified protocol.

Line 128, For cases vaginal sample and for control group vaginal and rectal swabs will be taken .Need justification.

In line 139, Outcomes will be censored at 42 days on which basis or reference ort justification needed.

In line 141, why “Conditional Logistic 142 regression models will be used to estimate the odds ratios of the associations between risk factors” –is not clear.

The statistical analysis plan also requires specified statistical tool or software to run the analysis is simply missing in this section.

Line 179-179, Sample size calculation should be mention as a particular section describing detail power of the test and proportions.

Discussion:

The discussion portion should be needed some comparison with others in the perspective of AMR and bacterial profile.

The Line 196 the second limitation of this study isn’t justified and it perhaps not the limitation regarding the comparative study.

The strength of the study lacks in the protocol version.

Final Comments:

Major Revision required

Reviewer #2: Dear Authors,

I believe the paper has its merits and is potentially useful for researcher concerning study protocol. As far as we know, a study protocol is an essential part of a research project. It is a document that describes the study in detail, including the background, rationale, objectives, design, methodology, statistical considerations, and organization of a clinical research study. The protocol acts as a ‘manual’ for members of the research team to ensure everyone adheres to the methods outlined. As the study gets underway, it can then be used to monitor the study’s progress and evaluate its outcomes. Thus, the authors have clearly demonstrated all the norms of a study protocol in right template that PLOSONE requested.

Following, I would like to review the main sections such as Abstract, Introduction, Material and Methods, Discussion

Abstract: should include the study registration number where applicable

Introduction: It seems that authors also give the rationale for the study. However, the way of demonstration has not been cleared enough (What is a design hypothesis to test, what new evidence it is anticipated to provide). Besides, should referred to the appropriate previous literature.

Material and Methods: the manuscript demonstrate the aim, design and setting study, inclusion and exclusion criteria.

For the sample size: It is necessary to calculate the sample size according to the WHO formula for case-control study, to see that it is appropriate to select 100-200 (case-controls). The authors also show that characteristics of participants and how sample will be selected. However, the description of processes, interventions and comparisons should be demonstrated clearer. Besides, when and how the outcomes will be measured need adding.

Other issues met the standard like data management plan, safety considerations, the type of data and statistical analyses planned.

Discussion:

The authors also address issues involved in performing the study that are not covered in other section like limitations of study design, dissemination plans.

Reviewer #3: Understanding Maternal Sepsis Risk Factors And Bacterial Etiology: A Case-Control Study Protocol

Summary of the Research

Maternal disorders are the third leading cause of sepsis globally, accounting for 5.7 million (12%) cases in 2017. There are increasing concerns about the emergence of antimicrobial resistance (AMR) in bacteria commonly causing maternal sepsis. Our objective was to describe the protocol for a clinical and microbiology laboratory study to understand the epidemiology and bacterial etiology of maternal sepsis in a tertiary Obstetrics and Gynaecology Hospital. This case-control study aims to recruit 100 cases and 200 controls at Tu Du Hospital in Ho Chi Minh City, Vietnam, which had approximately 55,000 births in 2022. Women aged ≥ 18 years and ≥ 28 weeks gestation having a singleton birth will be eligible for inclusion as cases or controls, unless they have an uncomplicated localised or chronic infection, or an infection with SARS-CoV-2 (COVID-19). Cases will include pregnant or recently pregnant women with sepsis recognised between the onset of labour and/or time of delivery/cessation of pregnancy for up to 42 days post-partum. Sepsis will be defined as suspected or confirmed infection with an obstetrically modified Sequential Organ Failure Assessment score of ≥ 2, treatment with intravenous antimicrobials and requested cultures of any bodily fluid. Controls will be matched by age, location, parity, mode of delivery and gestational age. Outcomes include risk factors associated with the development of maternal sepsis, the frequency of adverse outcomes due to maternal sepsis, bacterial etiology and AMR profiles of cases and controls.

Areas for improvement

Title:

Good title reflecting the content of the study.

Abstract:

The authors should keep the abbreviations to a minimum.

The authors should revise the language to improve readability.

The authors should remove Harari, Eastern Ethiopia from key words.

Introduction:

The authors should write about what you want the readers to know. Detail information should be written in introduction section.

The authors should mention the significant of the study.

The authors should revise introduction section for grammar issues and language to improve readability.

The authors should make sure that the abbreviation is within full form at the first time.

The authors should make this section more clear, so readers will understand what message you wanted them to understand.

Materials and methods Evaluation Design and Setting:

Clear section. However, the authors should mention the reasons for selecting the design and the setting for the study.

The authors should revise the language to improve readability.

Data analysis, clear.

Study population, clear

Ethical consideration, clear.

Discussion section, I think this is not discussion section as it is not yet this study conducted so It is better to change to conclusion section.

Limitation of the study, clear.

Overall: clear protocol, wish you all the best, just need to make sure from grammar and readability

References:

The authors should revise all references according to the guidelines provided.

Reviewer #4: The authors have chosen an important topic. It would be good to have some more description in how infections are managed currently in Vietnam and what are the most common pathogens. Are the standard protocols followed? In addition, it would be good to see some dummy tables for the data analysis that the authors will be using. What policy and program implications do the authors see when the study is complete?

7. PLOS authors have the option to publish the peer review history of their article (what does this mean?). If published, this will include your full peer review and any attached files.

Reviewer #1: **Yes: **Dr. Ummul Khair Alam

Reviewer #2: No

Reviewer #3: **Yes: **Zalikha Khamis Al-Marzouqi

Reviewer #4: No

---

## [Author Response · Author response to Decision Letter 0]

29 Apr 2024

Reviewer #1: 

Keyword:

Antimicrobial resistance should not be in the keyword as it’s not in line with title of the protocol

Response: Thank you for this comment. Antimicrobial resistance has been removed as a keyword and replaced with bacterial etiology. 

Abstract:

Line 30 contain was thus grammatical error. A protocol can’t contain was. Overall grammatical errors should be rewrite.

Response: Thank you for this comment. We have amended the wording as follows:

Our aim is to describe the protocol for a clinical and microbiology laboratory study to understand risk factors for and the bacterial etiology of maternal sepsis in a tertiary Obstetrics and Gynaecology Hospital. 

Line 43 primary and secondary outcome should be mentioned clearly in the abstract section

Response: Thank you for this comment, we have amended the wording as follows:

Primary and secondary outcomes are risk factors associated with the development of maternal sepsis, the frequency of adverse outcomes due to maternal sepsis, bacterial etiology and AMR profiles of cases and controls.

Overall aim and title conflict rigorously.

Response: Thank you for this comment, we have amended the wording in the abstract as follows:

Our aim is to describe the protocol for a clinical and microbiology laboratory study to understand risk factors for and the bacterial etiology of maternal sepsis in a tertiary Obstetrics and Gynaecology Hospital. 

Introduction:

Line 56 LMICs should be showed a grammatical mistake.

Response: Thank you for this comment, we have amended the acronym as requested to LMICs. 

Line 63, 64 should contain background data more to prove the statement as well.

Response: Thank you for this comment. We have added some additional context to the background paragraphs as follows:

Most efforts to tackle maternal sepsis have been integrated into broader initiatives to reduce maternal and neonatal mortality [7]. A major emphasis of programs has been on increasing the number of births with a skilled attendant, particularly within health facilities. It is estimated that three-quarters of all births globally now occur within health facilities, but as this number has increased, concerns regarding the quality of maternity care in health facilities have emerged [7]. 

Although antibiotics unquestionably save lives in childbirth [8], they should always be used in the context of antimicrobial stewardship programmes [7, 9]. Bacterial characterisation and antimicrobial susceptibility testing are important in determining appropriate treatment, particularly in LMICs where high rates of antimicrobial resistance (AMR) present one of the biggest challenges to global public health [9, 10]. Emerging concerns about antimicrobial resistance (AMR) in bacteria commonly causing maternal sepsis [7, 8] are polarised by an absence of data on antimicrobial stewardship and resistance in maternity units globally [7, 8]. 

Reducing maternal mortality and morbidity due to sepsis requires prevention, early diagnosis and prompt but appropriate management. Several interventions have been designed to prevent and promptly recognise sepsis, including early warning systems and care bundles. However, data describing how maternal sepsis is recognised in urban health care facilities are lacking [3], and there are no validated criteria for diagnosis.

Vietnam is the third largest LMIC in Southeast Asia with a population of close to 98.2 million [11]. Like other parts of Southeast Asia, Vietnam is a hotspot for emerging infectious diseases with severe dengue, Streptococcus and increased AMR listed as major causes of sepsis admissions in intensive care units [12]. While there is some evidence describing the country-specific etiology of sepsis and septic shock in Vietnam, these reports are limited to adults [13] and neonates [10, 14] requiring intensive care treatment. There are limited contemporary data on risk factors for and causes of sepsis, including the emergence of antimicrobial resistant bacteria pregnant women and women who have recently given birth [10]. 

Line 65 referencing should be corrected [7] [8]

Response: Thank you for this observation. We have amended the referencing as requested. 

More epidemiology on bacterial profile and AMR should be reflected in accordance with the study objective of the following concern study.

Response: Thank you for this comment. We hope we have satisfied the authors request in our answer provided above, specifically the additional information and context provided in the background section. 

Material and Methods:

Line 82 how cases and controls will be selected not clear. If same age and gestation will be treated as either case or control? Line 93 conflict with it without defining pregnant of which gestational age.

Response: Thank you for this comment regarding the presentation of our eligibility criteria. We have made the following changes to clarify the presentation of our eligibility criteria to improve readability: 

Inclusion criteria - cases

Cases will include pregnant or recently pregnant women with sepsis recognised between the onset of labour and/or time of delivery/cessation of pregnancy for up to 42 days post-partum. A diagnosis of sepsis will require the woman to have:

• Suspected or confirmed infection being treated with intravenous antibiotics 

• An Obstetrically Modified Sequential Organ Failure Assessment (SOFA) Score of ≥ 2 (Table 1)

• Requested cultures of any bodily fluid, including swab specimens

Inclusion criteria - controls

Controls will be pregnant or recently pregnant women who are up to 42 days post-partum who do not have sepsis. 

Controls will be matched by age (within 2-year age bands), location (urban vs rural), mode of delivery, parity (primiparous vs multiparous) and gestational age rounded to the closest week (where possible). For each case, two controls will be randomly selected to ensure the comparability of the controls to the general obstetric population.

Case and control Exclusion criteria 

1. Women aged < 18 years of age

2. Women who are < 28 weeks gestation 

3. Women who have an uncomplicated localised or chronic infection

4. Women who have an infection with SARS-CoV-2 (COVID-19) will be excluded.

Line 97 mode of delivery should be specified by NVD or CS 

Response: Thank you for this observation, we have added these options in brackets as follows: 

Controls will be matched by age (within 2-year age bands), location (urban vs rural), mode of delivery (normal vaginal delivery or caesarean section), parity (primiparous vs multiparous) and gestational age rounded to the closest week (where possible).

The Table -1, Obstetrically Modified SOFA score according to the reference [11] was not mentioned why respiratory parameter, coagulation and liver parameter was changed was not described and reference should be mentioned for the justified modification.

Response: Thank you for this comment, we have added the following paragraph and a further reference to justify the use of the chosen algorithm:

The Obstetrically Modified SOFA score has been used in accordance with recommendations from the Society of Obstetric Medicine in Australia and New Zealand [11]. In order to demonstrate evidence of organ dysfunction only a score of ≥2 is required [11]. To simplify the calculation for research coordinators who will be screening patients, we removed scores of 3 or 4 from the scoring sheet, as only a score of ≥2 is required for study inclusion. The modified SOFA parameters for pregnant or recently pregnant woman include a reduction in the creatinine threshold as serum creatinine levels are significantly reduced in pregnancy with normal ranges of 35-80 µmol/L, a simplified neurological assessment as Glasgow Coma Score is not routinely used on maternity wards. For this study, we also modified the respiratory SOFA score component, substituting the PaO2/FiO2 with the SPO2/FiO2 ratio [16], due to lack of availability of arterial blood gas monitoring in the hospital.

In line 117-118 how Bacterial etiology and AMR profile will be assessed through transmission dynamics or genomic analysis (16s r RNA extraction or shotgun metagenomics in which region v4 or any other should be mentioned clearly. The overall outcome should be focused and specified according to the research question and hypothesis that lack the specified protocol.

Response: The bacterial etiology and AMR profile are not assessed through transmission dynamics or genomic analysis. As detailed in the manuscript, the bacterial identification and susceptibility testing are standard procedures conducted at the hospital for all suspected sepsis cases: 

“Clinical samples will be collected and subjected to microbiological culture in accordance with routine clinical care and procedures. Where clinically indicated for patient management vaginal swabs will be collected. Bacterial identification and AMR testing will be performed in-house using the BD Phoenix™ automated identification and susceptibility testing system”

Whole genome sequencing and analysis will be performed for the same bacterial species identified from cases and controls to better understand what causes maternal sepsis. 16S or shotgun metagenomic sequencing will not be relevant for this study. 

Line 128, For cases vaginal sample and for control group vaginal and rectal swabs will be taken .Need justification.

Response: Thank you for this comment. For cases we are using swabs collected as per routine/standard of care which is usually vaginal swabs. We are taking a vaginal and rectal samples from controls to ensure we have a comprehensive profile of bacteria from women in the control group. This will ensure robustness of findings and strengthen comparisons made between the bacterial profiles of cases and controls. 

In line 139, Outcomes will be censored at 42 days on which basis or reference ort justification needed.

Response: Thank you for this comment, we have added the following end to the sentence to justify our choice of this timepoint. 

Outcomes will be censored at the end of the postpartum period (42 days) or at discharge, transfer to another treating facility or at time of death, whichever comes first.

In line 141, why “Conditional Logistic 142 regression models will be used to estimate the odds ratios of the associations between risk factors” –is not clear.

Response: Thank you for this comment, we have amended the sentence as follows:

Conditional logistic regression is to be used because we have a case-control study wherein the control to case ratio is not 1:1. We now say: 

Due to the uneven case-control distribution, and also due to the matched allocations, conditional logistic regression models will be used to compare the actions of putative risk factors between the cases and controls. 

The statistical analysis plan also requires specified statistical tool or software to run the analysis is simply missing in this section.

Response: Thank you for this comment. We have added the following at the end of the statistical analysis section:

Analyses will be conducted using Stata software.

Line 179-179, Sample size calculation should be mention as a particular section describing detail power of the test and proportions.

Response: Thank you for this comment. We have now added the following text: 

We chose to study 100 cases and 200 controls primarily because of the anticipated availability of suitable cases and controls within the limitations of the research environment. With this sampling allocation we will have 80% power, at the 5% significance level and using a two-sided test, to detect an odds ratio of 3, or less, whenever the risk factor has a prevalence of 8%, or more.

Discussion:

The discussion portion should be needed some comparison with others in the perspective of AMR and bacterial profile.

Response: Thank you for this comment. As we are reporting a protocol for a planned study, rather than the results of an actual study, we have intentionally kept the discussion section succinct but would plan to report a comparison to other studies in our results when we publish them. 

The Line 196 the second limitation of this study isn’t justified and it perhaps not the limitation regarding the comparative study.

The strength of the study lacks in the protocol version.

Response: Thank you for these comments related to the strengths and limitations of the study protocol. As we are reporting a protocol for a planned study, rather than the results of an actual study, we believe that the strengths and limitations reported are appropriate and relevant for a paper of this type (protocol paper). 

Final Comments:

Major Revision required

Response: Thank you for your time in reviewing our paper.

Reviewer #2: Dear Authors,

I believe the paper has its merits and is potentially useful for researcher concerning study protocol. As far as we know, a study protocol is an essential part of a research project. It is a document that describes the study in detail, including the background, rationale, objectives, design, methodology, statistical considerations, and organization of a clinical research study. The protocol acts as a ‘manual’ for members of the research team to ensure everyone adheres to the methods outlined. As the study gets underway, it can then be used to monitor the study’s progress and evaluate its outcomes. Thus, the authors have clearly demonstrated all the norms of a study protocol in right template that PLOSONE requested.

Response: Thank you for this feedback. 

Following, I would like to review the main sections such as Abstract, Introduction, Material and Methods, Discussion

Abstract: should include the study registration number where applicable

Response: Thank you for this feedback. As the study is not a clinical trial we have not registered the study and are not aware of any websites where we can register a case-control study. If the reviewer does know of a site for registering case control studies, please let us know and we will update the manuscript accordingly. 

Introduction: It seems that authors also give the rationale for the study. However, the way of demonstration has not been cleared enough (What is a design hypothesis to test, what new evidence it is anticipated to provide). Besides, should referred to the appropriate previous literature.

Response: Thank you for this feedback. We have added some background information and further rationale for conducting this study in response to a similar comment from reviewer 1, please see our full response provided above and in tracked changes in the updated manuscript background section. 

Material and Methods: the manuscript demonstrate the aim, design and setting study, inclusion and exclusion criteria.

Response: Thank you for this feedback. As per our response to Reviewer 1, we have updated the study methods and inclusion criteria for clarity to read as follows: 

Inclusion criteria - cases

Cases will include pregnant or recently pregnant women with sepsis recognised between the onset of labour and/or time of delivery/cessation of pregnancy for up to 42 days post-partum. A diagnosis of sepsis will require the woman to have:

• Suspected or confirmed infection being treated with intravenous antibiotics 

• An Obstetrically Modified Sequential Organ Failure Assessment (SOFA) Score of ≥ 2 (Table 1)

• Requested cultures of any bodily fluid, including swab specimens

Inclusion criteria - controls

Controls will include pregnant or recently pregnant women who are up to 42 days post-partum who do not have sepsis. 

Controls will be matched by age (within 2-year age bands), location (urban vs rural), mode of delivery, parity (primiparous vs multiparous) and gestational age rounded to the closest week (where possible). For each case, two controls will be randomly selected to ensure the comparability of the controls to the general obstetric population.

Case and control Exclusion criteria 

5. Women aged < 18 years of age

6. Women who are < 28 weeks gestation 

7. Women who have an uncomplicated localised or chronic infection

8. Women who have an infection with SARS-CoV-2 (COVID-19) will be excluded.

For the sample size: It is necessary to calculate the sample size according to the WHO formula for case-control study, to see that it is appropriate

---

## [Decision Letter · Decision Letter 1]

30 May 2024

Understanding maternal sepsis risk factors and bacterial etiology: a case-control study protocol

PONE-D-23-40494R1

Dear Dr. Kelly Thompson,

We’re pleased to inform you that your manuscript has been judged scientifically suitable for publication and will be formally accepted for publication once it meets all outstanding technical requirements.

Kind regards,

Trung Quang Nguyen

Academic Editor

PLOS ONE

Additional Editor Comments (optional):

Reviewers' comments:

Reviewer's Responses to Questions

**Comments to the Author**

1. Does the manuscript provide a valid rationale for the proposed study, with clearly identified and justified research questions?

Reviewer #3: Yes

2. Is the protocol technically sound and planned in a manner that will lead to a meaningful outcome and allow testing the stated hypotheses?

Reviewer #3: Yes

3. Is the methodology feasible and described in sufficient detail to allow the work to be replicable?

Reviewer #3: Yes

4. Have the authors described where all data underlying the findings will be made available when the study is complete?

Reviewer #3: Yes

5. Is the manuscript presented in an intelligible fashion and written in standard English?

Reviewer #3: Yes

6. Review Comments to the Author

You may also provide optional suggestions and comments to authors that they might find helpful in planning their study.

Reviewer #3: UNDERSTANDING MATERNAL SEPSIS RISK FACTORS AND BACTERIAL ETIOLOGY: A CASE-CONTROL STUDY PROTOCOL

Summary of the Research Protocol

This study objective was to describe the protocol for a clinical and microbiology laboratory study to understand the epidemiology and bacterial etiology of maternal sepsis in a tertiary Obstetrics and Gynaecology Hospital. Case-control study aims to recruit 100 cases and 200 controls at Tu Du Hospital in Ho Chi Minh City, Vietnam, which had approximately 55,000 births in 2022. Women aged ≥ 18 years and ≥ 28 weeks gestation having a singleton birth will be eligible for inclusion as cases or controls, unless they have an uncomplicated localised or chronic infection, or an infection with SARS-CoV-2 (COVID-19). Cases will include pregnant or recently pregnant women with sepsis recognised between the onset of labour and/or time of delivery/cessation of pregnancy for up to 42 days post-partum. Sepsis will be defined as suspected or confirmed infection with an obstetrically modified Sequential Organ Failure Assessment score of ≥ 2, treatment with intravenous antimicrobials and requested cultures of any bodily fluid. Controls will be matched by age, location, parity, mode of delivery and gestational age. Outcomes include risk factors associated with the development of maternal sepsis, the frequency of adverse outcomes due to maternal sepsis, bacterial etiology and AMR profiles of cases and controls. Implicational of the study: The study will improve understanding of the epidemiology and clinical implications of maternal sepsis management including the presence of AMR in women giving birth in Vietnam. It will help us to determine whether women in this setting are receiving optimal care and to identify opportunities for improvement.

Areas for improvement

Title:

Concise reflecting the aim of the study, well written.

Abstract:

Well written.

Introduction:

Well done, well written introduction section; however, the authors should revise introduction section for grammar issues and language to improve readability.

Material and Methods:

Well written material and methods however, the authors need not to use words like “we”, this is not academic words. Use words like the authors will….

Discussion:

Well done, clear.

References:

The authors should revise all references according to the guidelines provided.

Overall, well written and clear protocol. All the best

7. PLOS authors have the option to publish the peer review history of their article (what does this mean?). If published, this will include your full peer review and any attached files.

Reviewer #3: **Yes: **Zalikha Khamis Al-Marzouqi

---

## [Editor Report · Acceptance letter]

17 Jun 2024

PONE-D-23-40494R1 

PLOS ONE

Dear Dr. Thompson, 

I'm pleased to inform you that your manuscript has been deemed suitable for publication in PLOS ONE. Congratulations! Your manuscript is now being handed over to our production team.

Kind regards, 

on behalf of

Dr. Trung Quang Nguyen 

Academic Editor

PLOS ONE